# CHELSA-W5E5: Daily 1 km meteorological forcing data for climate impact studies

Dirk Nikolaus Karger[1], Stefan Lange[2], Chantal Hari[1,3], Christopher P.O. Reyer[2], Olaf Conrad[4], Niklaus E. Zimmermann[1], Katja Frieler[2]

[1] Swiss Federal Research Institute WSL, Zürcherstrasse 111, 8903 Birmensdorf, Switzerland
[2] Potsdam Institute for Climate Impact Research (PIK), Member of Leibniz Association, P.O.Box 601203, 14412, Potsdam, Germany
[3] Wyss Academy for Nature at the University of Bern, Kochergasse 4, 3011 Bern, Switzerland
[4] University of Hamburg, Bundesstraße 55, 20146 Hamburg, Germany

*Correspondence to: Dirk Nikolaus Karger (dirk.karger@wsl.ch)*

**Abstract.** Current changes in the world's climate increasingly impact a wide variety of sectors globally, from agriculture, ecosystems, to water and energy supply or human health. Many impacts of climate on these sectors happen at high spatio-temporal resolutions that are not covered by current global climate datasets. Here we present CHELSA-W5E5 (https://doi.org/10.48364/ISIMIP.836809.3, Karger et al., 2022): a climate forcing dataset at daily temporal resolution and 30 arcsec spatial resolution for air-temperatures, precipitation rates, and downwelling shortwave solar radiation. This dataset is a spatially downscaled version of the 0.5° W5E5 dataset using the CHELSA V2 topographic downscaling algorithm. We show that the downscaling generally increases the accuracy of climate data by decreasing the bias, and increasing the correlation with measurements from meteorological stations. Bias reductions are largest in topographically complex terrain. Limitations arise for minimum near surface air temperatures in regions that are prone to cold air pooling, or at the upper extreme end of surface downwelling shortwave radiation. We further show that our topographically downscaled climate data compare well with the results of dynamical downscaling using the regional climate model WRF, as time series from both sources are similarly well correlated to station observations. This is remarkable given the lower computational cost of the CHELSA V2 algorithm compared to WRF and similar models. Overall, we conclude that the downscaling can provide higher resolution climate data with increased accuracy. Hence, the dataset will be of value for a wide range of climate change impact studies both at global level but also as for applications that cover more than one region and benefit from using a consistent dataset across these regions.

## 1 Introduction

With ongoing climate change, the assessment of climate change impacts on natural and social systems requires increasing attention (IPCC, 2022). Historically, a strong focus has been on the scientific exploration of climate impacts on agriculture, forestry, water management, human health, and other sectors by using climate impact models driven by historical or projected future climate data. Yet, with observed climate change impacts emerging widely already at current levels of warming (IPCC, 2022),  a wide range of decision-making processes as well as business activities increasingly rely on actionable knowledge from impact models that is useful beyond the scientific community which is developing them. For example, so-called climate services are designed to support adaptation of stakeholders and their activities in response to climate change (Brasseur and Gallardo, 2016; Hewitt et al., 2012; Lourenço et al., 2016), where the attribution of climate impacts has become highly relevant for climate litigation (Mengel et al., 2021). There is also an increasing demand to quantify damages that cannot be avoided by climate mitigation or adaptation (Huber et al., 2022). These activities require highly accurate climate impact datasets at high spatio-temporal resolution. Daily temporal resolution for example allows capturing extreme events such as heavy precipitation or heat waves that would not be visible at monthly resolution (Ban et al., 2021). Likewise, high spatial resolution (e.g. 30 arcsec, i.e. ~1 km at the equator) allows to capture topographic effects in mountainous areas or patterns of climate variables with small-scale spatial variability (Gerlitz et al., 2015; Daly et al., 1994).

High resolution climate data can typically be produced using either regional climate models for dynamical downscaling (Giorgi et al., 2009), statistical downscaling methods using large-scale predictors of the small-scale state of the atmosphere (Maraun and Widmann, 2018), or topographic downscaling methods that mainly use terrain based predictors to increase the spatial resolution of climate data (Karger et al., 2017; Fiddes and Gruber, 2014). Regional climate models have the advantage of representing the fundamental physical, chemical and biological processes of the climate system. While this makes them powerful tools for studying future climates it also makes them computationally expensive wherefore they cannot easily be applied at the global level (Giorgi et al., 2009; Sørland et al., 2021; Schär et al., 2019). Statistical downscaling methods are based on empirical relationships between large-scale predictors and small-scale predictands (Wilby et al., 1998). These relationships are typically derived from historical observations of predictors and predictands and then applied to downscale large-scale climate projections. While this is computationally less expensive, it implies out-of-sample applications of a statistical model, which may lead to physically implausible results (Maraun et al., 2017; Lanzante et al., 2018). Lastly, topographic downscaling methods primarily use terrain based information to add small-scale details to large-scale inputs, such as the influence of mountain ranges on precipitation patterns (Roe, 2005). Examples of such methods include Climatologies at high resolution for the Earth's land surface areas (CHELSA) (Karger et al., 2017, 2021, 2020), and the Parameter-elevation Regressions on Independent Slopes Model (PRISM) (Daly et al., 1997, 1994). Considering the out-of-sample limitation of statistical downscaling, topographic downscaling of climate projections is less problematic in comparison, at similar computational cost. On the downside, topographic downscaling is based on mechanistic equations which, due to their simplicity, may still introduce biases in the climate data (Karger et al., 2017, 2021). In addition, those equations are unable to represent small-scale spatial patterns that are unrelated to topography, such as small-scale convective precipitation over flat terrain (Karger et al., 2021).

All approaches have historically been challenged by computational and storage limitations if carried out at the global level (Schär et al., 2019). For example, the latest global reanalysis dataset based on the dynamic land surface model 'Hydrology in the Tiled ECMWF Scheme for Surface Exchanges over Land' (HTESSEL, Balsamo et al., 2009) is only available at a resolution of ~9 km which still masks important local climate variability (Muñoz-Sabater et al., 2021). For these reasons, climate datasets at high spatial and temporal solution usually only exist at local to regional levels, which is adequate for analyses at these levels. However, there are no global products representing temperature, solar radiation, and precipitation at both high temporal (daily) and high spatial (~1 km) resolution, although these would offer considerable benefits to climate impact modelling. For example, a consistent global dataset that allows regional hydrological models to be run at various locations using consistent climate driving data so that impacts can be integrated across regions (Huang et al., 2017; Krysanova and Hattermann, 2017). Likewise, global analyses that are strongly dependent on the resolution of the data could be carried out at much finer resolution than is currently the case. For example, Shi et al., (2021) calculated how aridity velocity affects a wide range of species using climate data at 0.5 degree resolution, yet this resolution neglects important topographic details that are important as species might benefit from topographic diversity for surviving extreme climatic conditions (Barton et al., 2019).

To address this gap in data availability and to enable tests of how beneficial such global datasets would be, the objective of this paper is to present a global climate dataset at 30 arcsec and daily resolution: CHELSA-W5E5 v1.0 (https://doi.org/10.48364/ISIMIP.836809.3, Karger et al., 2022). This dataset builds upon WFDE5 over land merged with ERA5 over the ocean (W5E5) v1.0, an observational climate dataset that has been thoroughly evaluated and intensively used in climate impact modelling (Lange, 2019; Cucchi et al., 2020). CHELSA-W5E5 v1.0 is derived from W5E5 via topographic downscaling using the CHELSA V2 algorithm (Karger et al., 2017, 2020, 2021). Through a detailed evaluation of CHELSA-W5E5 v1.0, we aim to demonstrate the added value of a kilometre-scale resolution downscaling compared to the coarse resolution (0.5°) W5E5 data. We focus on a set of key climatic variables that are high relevant for climate impact modelling, namely daily minimum (*tasmin*, in units of K), mean (*tas*, K) and maximum (*tasmax*, K) near-surface (2 m) air temperature, which are, for example, relevant for assessing heat extremes (Huber et al., 2020), daily mean precipitation rate (*pr*, kg m$^{-2}$ s$^{-1}$), a crucial variable for example for hydrological and vegetation models (Müller Schmied et al., 2014; Chang et al., 2017), as well as daily mean surface downwelling shortwave radiation (*rsds*, W m$^{-2}$), which is for example crucial for agricultural modelling (Ruane et al., 2015, 2021). The analyses and data are building on earlier efforts to downscale precipitation (Karger et al., 2017, 2020, 2021) and we focus on assessing where the new dataset improves the estimate of a climate variable by moving to a spatial resolution of 30 arcsec, and what caveats have to be kept in mind when applying the data for climate impact analyses.

Here we describe the CHELSA downscaling procedure applied to W5E5 and evaluate its performance in improving the accuracy of modelled air-temperatures, precipitation rates, and downwelling shortwave solar radiation. We give a description on the input data as well as a detailed description of the downscaling procedure applied, which includes the downscaling of near-surface air temperature (*tas*, *tasmax*, *tasmin*), surface downwelling shortwave radiation (*rsds*), and precipitation (pr). We evaluate our results using observations at meteorological stations, and analyse the performance of the downscaling globally, regionally, and seasonally, as well as at the extremes and additionally compare our results with dynamically downscaled data.

## 2. Material and Methods

To downscale the coarse resolution W5E5 data we used the CHELSA V2 algorithm (Karger et al., 2017). This algorithm is a topographically informed, mechanistic downscaling method. It downscales 2m air-temperatures (*tas, tasmax, tasmin*) based on air temperature lapse rates in the lower atmosphere, precipitation rates (*pr*) using orographic terrain effects, and surface downwelling shortwave radiation (*rsds*) using a mechanistic terrain-based downscaling. In the following we describe the input data and downscaling procedure in more detail which is shown in Fig. 1.

### 2.1. Input data

### 2.1.1 W5E5

WFDE5 over land merged with ERA5 over the ocean (W5E5) v1.0 (Lange, 2019) is the observational reference climate input dataset used in the Inter-Sectoral Impact Model Intercomparison Project phase 3 (ISIMIP3, www.isimip.org). It covers the years 1979-2016 for the entire globe. The data have daily temporal and 0.5° spatial resolution. W5E5 combines the Waterer and global Change (WATCH) Forcing Data methodology applied to ERA5 reanalysis data (WFDE5) v1.0 (Cucchi et al., 2020) over land with data from the latest version of the European Reanalysis (ERA5) (Hersbach et al., 2020) over the ocean. In the following we briefly describe ERA5, WFDE5 and W5E5.

The ERA5 global reanalysis (Hersbach et al., 2020) is produced at the European Centre for Medium Range Weather Forecast (ECMWF) as part of the EU-funded Copernicus Climate Change Service (C3S). It is the successor of ERA-Interim (Dee, 2011) and in comparison benefits from 10 years of developments of the underlying weather forecast model and data assimilation system. More observations are assimilated in ERA5 than in its predecessor ERA-Interim, including stratospheric sulphate aerosols. In addition, ERA5 has higher temporal and spatial resolution (hourly and 0.25° compared to 3-hourly and 0.7°).

The WFDE5 meteorological forcing dataset is a bias-adjusted version of ERA5 that covers the global land surface at hourly temporal and 0.5° spatial resolution for selected near-surface atmospheric variables (air temperature, shortwave and longwave downwelling radiation, rainfall and snowfall, specific humidity, air pressure, and wind speed). Bias adjustments were applied according to the WATCH Forcing Data methodology (Weedon et al., 2014, 2011). That means that (i) monthly mean values of daily mean temperature and the diurnal temperature range were elevation-adjusted and bias-adjusted using version 4.03 of the Climate Research Unit gridded Time Series (CRU TS) (Harris et al., 2020), (ii) pressure, humidity and longwave radiation were aligned with the adjusted temperature, (iii) monthly mean shortwave radiation was bias-adjusted using aerosol correction factors (Cucchi et al., 2020) and CRU TS4.03 cloud cover, and (iv) rainfall and snowfall rates were bias-adjusted with respect to the monthly number of wet days using CRU TS4.03, monthly precipitation totals using observations from either CRU TS4.03 or data from the Global Precipitation Climatology Centre (GPCC) full data product version 2018 (Schneider et al., 2018), followed by a gauge-catch correction, and a correction of the snowfall-to-rainfall ratio using the adjusted temperature (Cucchi et al., 2020). Using either CRU TS4.03 or GPCCv2018 precipitation totals, two different WFDE5 precipitation datasets were produced. The variant based on GPCCv2018 was used for W5E5. Wind speed is the only variable that was not adjusted.

Lastly, W5E5 combines WFDE5 data over land with ERA5 data over the ocean to cover the whole globe at daily temporal and 0.5° spatial resolution. Here we use daily total precipitation (*pr*), daily mean downwelling shortwave

radiation (*rsds*) as well as daily mean, minimum and maximum near-surface air temperature (*tas, tasmin* and *tasmax*, respectively) from W5E5. The daily temperature values are equal to the daily mean (for *tas*), minimum (for *tasmin*) and maximum (for *tasmax*) of the hourly temperature values from WFDE5 over land and ERA5 aggregated to 0.5° spatial resolution over the ocean. Similarly, W5E5 *pr* (*rsds*) is equal to the daily sum (mean) of hourly total precipitation (shortwave radiation) from WFDE5 over land and ERA5 aggregated to 0.5° spatial resolution over the ocean, with the following exception: W5E5 *pr* over the ocean was bias-adjusted using monthly precipitation totals from version 2.3 of the Global Precipitation Climatology Project (Adler et al., 2003). Monthly rescaling factors used for this purpose were computed following the scale-selective rescaling procedure described by (Balsamo et al., 2010).

### 2.1.2 Global Multi-resolution Terrain Elevation Data 2010 (GMTED2010)

The Global Multi-resolution Terrain Elevation Data 2010 (GMTED2010) (Danielson and Gesch, 2011) dataset contains elevation data for the globe collected from various sources at resolutions from 7.5 arcsec to 30 arcsec. We use the 30 arcsec version of the data that represents the mean elevation of all 7.5 arcsec grid cells.

### 2.1.3 Land-sea mask

The CHELSA downscaling algorithm only has an effect where topography varies in space. Over the ocean, the output of the downscaling is equivalent to a simple B-spline interpolation of the input data. To reduce the size of the high-resolution dataset, we therefore applied a land-sea mask that is intended to cut out the parts over the ocean that are not affected by topography. To make sure this mask actually covers all land masses, a cell of the 30 arcsec CHELSA-W5E5 grid is considered a land grid cell if it overlaps with any of the land polygons provided by the global, self-consistent, hierarchical, high-resolution shoreline database (GSHHG) v2.3.7 (Wessel and Smith, 1996), the 30-m spatial resolution global shoreline vector (GSV) (Sayre et al., 2019), and the MODIS-based Mosaic of Antarctica data (MOA) (Scambos et al., 2007). To ensure all land pixels are covered we additionally added a buffer of 60 arcsec width to the boundaries of each land polygon.

## 2.2 Downscaling procedure

### 2.2.1 Downscaling of near-surface air temperature (*tas, tasmax, tasmin*)

The CHELSA downscaling algorithm was applied day by day. The downscaling of W5E5 air temperature (*tas, tasmax, tasmin*) was done by using a daily mean near-surface atmospheric temperature lapse rate, $\overline{\Gamma}$, derived from ERA5, combined with differences in surface altitude between GMTED2010 and W5E5. Here, $\overline{\Gamma}$ is the daily mean of hourly lapse rates, $\Gamma$, with

$$\Gamma = (t_{850} - t_{950}) / (z_{850} - z_{950}), \tag{1}$$

where $t_{850}$ and $t_{950}$ are ERA5 hourly air temperatures at 850 hPa and 950 hPa, respectively, and $z_{850}$ and $z_{950}$ are the geopotential heights of those pressure levels multiplied by the gravitational constant (9.80665 m s$^{-2}$*).* We then interpolated W5E5 *tas, tasmax* and *tasmin* from their original resolution of 0.5° to the 30 arcsec resolution of GMTED2010 using a B-spline interpolation (see Karger et al., 2021 for an example on how the B-Spline interpolation is implemented), resulting in an interpolated high-resolution temperature surface, $t_c$. To include the

high-resolution topography, we first interpolated the 0.5° orography from W5E5 to 30 arcsec using a B-spline interpolation, this way creating a reference elevation grid, $z_c$, that corresponds to $t_c$. We then used $\bar{\Gamma}$ together with $z_c$ and $z_h$, the GMTED2010 orography at 30 arcsec, to do the topographic downscaling of $t_c$, according to

$$t_h = t_c + \bar{\Gamma} \cdot ( z_h - z_c ),$$ (2)

5 where $t_h$ is the downscaled near-surface air temperature at 30 arcsec resolution, either being *tas*, *tasmax*, or *tasmin*.

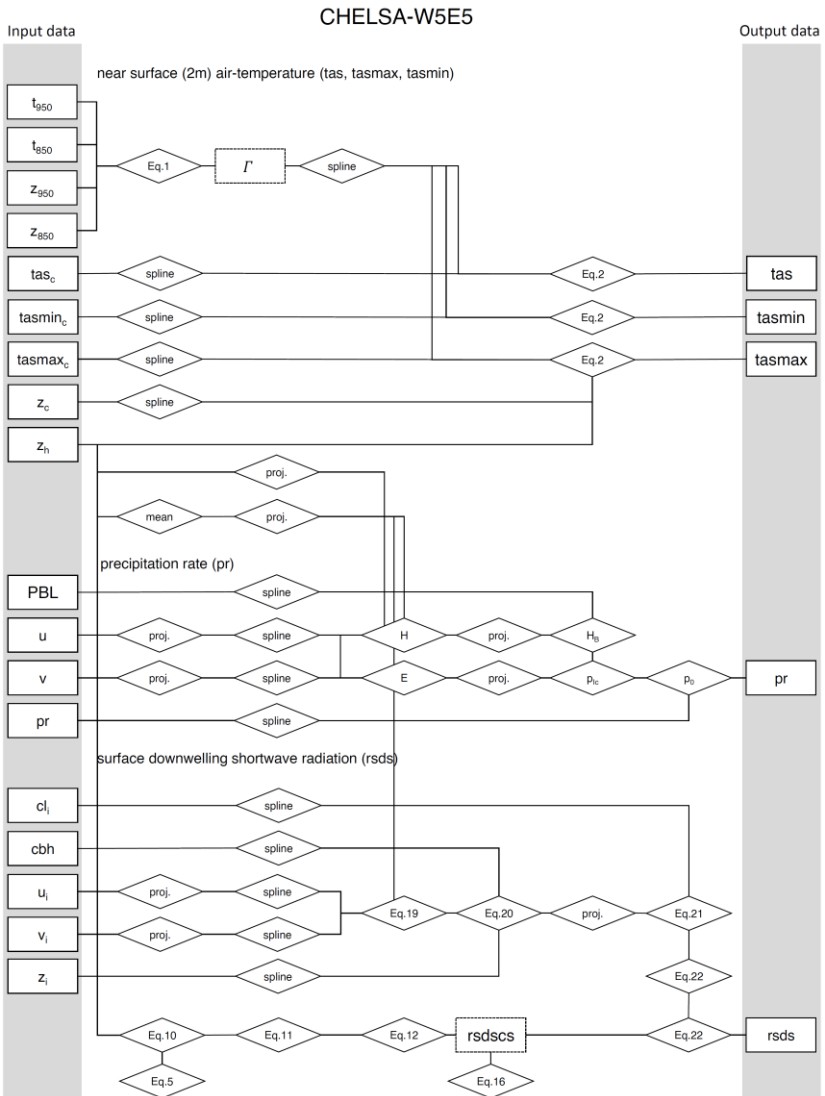

**Figure 1:** Schematic representation of the most important calculation steps (rhombi), input, and output data (rectangles) of the CHELSA-W5E5 downscaling. Intermediate data that is not part of the published data (temperature lapse rate; $\Gamma$ and clear sky solar radiation; *rsdscs*). Only the most important equations are indicated. Spline indicates that a B-spline interpolation is used to change the spatial resolution to a higher target resolution. Mean indicates that the mean across grid cells is used. Proj. indicates that a reprojection to another geographic projection is performed. For the respective abbreviations see the equations in the main text.

**2.2.2 Downscaling of surface downwelling shortwave radiation (*rsds*)**

Surface downwelling shortwave radiation at 30 arcsec resolution is strongly influenced by topographic features such as aspect or terrain shadows that are less pronounced at 0.5° resolution. The CHELSA downscaling algorithm combines such geometric effects with orographic effects on cloud cover for a topographic downscaling of *rsds*.

Geometric effects are considered by computing 30-arcsec clear-sky radiation estimates using the methods described in Böhner and Antonic (2009) as well as Wilson and Gallant (2000). This approach assumes that the net shortwave radiation, $S_n$, can be expressed as

$$S_n = S_s + S_h + S_t - S_r = (S_s + S_h + S_t) \cdot (1 - r), \tag{3}$$

with $S_n$ being the sum of all direct solar radiation received from sun, $S_s$, diffuse solar radiation received from the sky's hemisphere, $S_h$, radiation by reflection of surrounding land surfaces, $S_t$, minus the radiation which is reflected off the surface, $S_r$. Alternatively, the reflected fraction of the incoming radiation can be expressed using the dimensionless surface albedo, $r$. This formula for $S_n$ is strictly only valid for a horizontal, unobstructed surface. However, topography can severely influence net shortwave solar radiation by e.g. shading. A topographically corrected $S_n$, $S_n^*$, is given by

$$S_n^* = (S_s^* + S_h^* + S_t) \cdot (1 - r), \tag{4}$$

where $S_s^*$ and $S_h^*$ are direct and diffuse solar radiation modified by the surrounding topography of a given 30 arcsec grid cell and $S_t$ gives the reflection from surrounding land surfaces.

### 2.2.3 Direct solar radiation under clear sky conditions

Topographic direct solar radiation $S_s^*$ is calculated using:

$$\sin \theta = \cos \lambda \cos \delta \cos \varpi + \sin \lambda \sin \delta, \tag{5}$$

$$\cos \varphi = \frac{\cos \delta \cos \varpi - \sin \theta \cos \lambda}{\sin \lambda \cos \theta}, \tag{6}$$

$$\delta = 23.45 \cdot \sin \left( \frac{360° \cdot [284 + J])}{365} \right), \tag{7}$$

$$\varpi = 15° \cdot (12 - h), \tag{8}$$

where $\theta$ is the sun elevation angle, $\varphi$ is sun azimuth, $\lambda$ is the latitude, $\delta$ is the solar declination angle, $J$ is Julian day number, $\varpi$ is the hour angle in degrees, and the value $12 - h$ is equal to the distance of the given mid-hour from the true solar noon (0.5, 1.5, 2.5 h, etc.).

The angle between a plane orthogonal to sun's rays and terrain (solar illumination angle, γ) is calculated at time steps of 15 minutes using

$$\cos \gamma = \cos \beta \cdot \sin \theta + \sin \beta \cdot \cos \theta \cdot \cos(\varphi - \alpha), \tag{9}$$

where $\beta$ and $\alpha$ are surface slope and aspect, respectively, calculated from the high-resolution orography, $z_h$, and $\theta$ and $\varphi$ define the sun position on the sky. Shadowing from topography is calculated using the horizon angle, $\varphi$, which is defined as the maximum angle toward any other point in a given azimuth within 10,000 m horizontal distance,

$$\varphi = max_{d \leq 10,000 \, m} \, arctan \left( \frac{\Delta z(d)}{d} \right), \tag{10}$$

where $d$ is the distance to the point with higher elevation *and $\Delta z(d)$* is the associated elevation difference. Topographic direct radiation at hour $h$, $S_s^*(h)$, is then calculated using

$$S_s^*(h) = \varsigma(h) \frac{S_s(h)}{\sin \theta} \cos \gamma , \tag{11}$$

where $\varsigma(h)$ indicates if a terrain shadow is present (with $\varsigma(h)=0$ representing shadow and $\varsigma(h)=1$ representing no shadow), depending on $h$ and the horizon angle, $S_s(h)$ is the direct solar radiation on an unobstructed horizontal surface at hour $h$, and $\theta$ and $\gamma$ also depend on $h$ via their dependence on $\varpi$. The inclusion of the effect of terrain angle, is done by the division using $\sin \theta$ that tilts the horizontal surface to a surface that is orthogonal to the sun's rays. Multiplication by $\cos \gamma$ accounts for terrain. $S_s(h)$ also depends on the structure and the composition of the atmosphere. We assume a homogenous atmosphere with a transmissivity $\tau$ of 80% and then calculate $S_s(h)$ following (Wilson and Gallant, 2000) using

$$S_s(h) = \sin \theta \, G_{SC} \tau^m, \tag{12}$$

where $G_{SC}$ is the solar constant defined at 1367 kW m$^2$, and $m$ is the optical air mass, i.e. the length of the atmospheric path transversed by the sun's rays (List, 1968). For a sun elevation angle $\theta > 30°$, $m$ is calculated following (Linacre, 1992) using

$$m = \frac{1}{\cos(90-\theta)}, \tag{13}$$

and for $\theta \leq 30°$ the optical air mass $m$ is determined in 1 degree increments from a vector of known values after (List, 1968, p. 422), by using increments of 1° where:

$M$ = {2.00, 2.06, 2.12, 2.19, 2.27, 2.36, 2.45, 2.55, 2.65, 2.77, 2.90, 3.05, 3.21, 3.39, 3.59, 3.82, 4.07, 4.37, 4.72, 5.12, 5.60, 6.18, 6.88, 7.77, 8.90, 10.39, 12.44, 15.36, 19.79, 26.96, 26.96, 26.96}

Then m is calculated using element i in M, where i is the position of θ in M, by:

$$m = M_i + (\theta - i) * (M_{i+1} - M_i) \tag{14}$$

Daily mean topographic direct radiation, $\overline{S_s^*}$, is obtained via integration over all 15 minute time steps of the day,

$$\overline{S_s^*} = \frac{1}{n}\sum_{h=1}^{n} S_s^*(h) = \frac{1}{n}\sum_{h=1}^{n} \varsigma(h) \frac{S_s(h)}{\sin \theta} \cos \gamma, \tag{15}$$

where $n$ denotes the number of 15 min intervals of the day.

### 2.2.4 Diffuse solar radiation under clear sky conditions

Topographic corrected diffuse radiation $S_h^*$ is calculated by quantifying how much of the sky is visible from a grid cell, using

$$S_h^* = S_h \Psi_s, \tag{16}$$

where $\Psi_s$ is based on the horizon angles $\phi_i$ in different azimuth directions $\Phi_i$ of the full circle originating in a focal grid cell, and $S_h$ being the diffuse solar radiation calculated using:

$$S_h = (0.271 - 0.294\,\tau^m)G_{SC} \cdot \Psi_s, \tag{17}$$

Where $G_{SC}$ is the solar constant defined at 1367 kW m², and $\Psi_s$ is the sky view factor defined as

$$\Psi_s = \frac{1}{N}\sum_{i=1}^{N}\ [\cos\beta \cos\varphi_i + \sin\beta \cos(\Phi_i - \alpha)\ \cdot\ (90 - \varphi_i - \sin\varphi_i \cos\varphi_i)\ ], \tag{18}$$

with $N=8$ uniformly distributed directions used for an approximation of the topographic effect.

## 2.2.5 shortwave downwelling solar radiation under cloudy conditions

To calculate *rsds* under cloudy conditions, we calculated surface cloud area fraction (*clt*) from atmospheric cloud fractions *cl* at pressure levels *z* from ERA5. We first calculated the windward leeward index *H* using the *u* and *v* wind components from ERA5 following the methods described in Karger et al 2021. To distinguish between clouds that are influenced by orography from clouds in the free atmosphere, we first adjusted the windward leeward index relative to the number of pressure levels used, so that the windward leeward index is stronger at lower pressure levels than on pressure levels that are not influenced by the orography anymore. For each pressure level $i..n$ we calculated the corrected windward leeward index $H_i^{cor1}$ using

$$H_i^{cor1} =\ H_i + (1 - H_i)\cdot\frac{i}{n-1}. \tag{19}$$

This gives however the highest orographic effect directly at the surface altitude *z*, where often cloud formation is not possible yet. We therefore additionally corrected the windward leeward index by its distance to the cloud base height derived from its altitude $z_i$ , and then B-spline interpolated to the 30 arcsec resolution *cbh* using,

$$H_i^{cor2} =\ H_i^{cor1} -\ (1 - H_i^{cor1})\cdot\frac{z_i - cbh}{cbh}, \tag{20}$$

where the cloud area fraction on each pressure level *i* is then given by a horizontal spline interpolation of the coarse grid cloud fraction to a 30 arcsec resolution $cl_i^c$ with the corrected windward leeward index:

$$cl_i^h = H_i^{cor2}\cdot S(cl_i^c) \tag{21}$$

Cloud area fraction at the ground level then follows the maximum overlap assumption so that:

$$clt\ =\ max(cl_1^h...cl_i^h) \tag{22}$$

To include surface cloud area fraction *clt* in *rsds* we used the parametrization from (Kasten and Czeplak, 1980):

$$rsds = S_n^*(1 - 0.75\cdot clt^{3.4}) \tag{23}$$

### 2.2.6 Downscaling of precipitation (pr)

The downscaling method for precipitation mostly follows that of Karger et al. (2021) but does not include the cloud cover correction based on satellite observations as those are not available for all years. We used the zonal

and meridional wind components as well as the height of the planetary boundary layer to calculate the windward leeward index $H$. $H$ together with the height of the boundary layer following Karger et al. (2021) were used for a first approximation of the orographic precipitation intensity, $H \approx p_i$, for the 30 arcsec resolution grid cell $i$. We then used a linear relationship between the input precipitation rate from W5E5, $pr_{W5E5}$, and $p_i$ to compute the downscaled precipitation of grid cells $I$, $pr_i$, according to

$$pr_i = \frac{p_i}{\frac{1}{n}\sum_{i=1}^{n} p_i} * pr_{W5E5}, \tag{24}$$

where $n$ equals the number of 30 arcsec grid cells that fall within a 0.5° grid cell. This equation ensures that the data are to scale, i.e., the precipitation flux at 0.5° resolution is preserved. More details on the exact parametrization of the downscaling algorithm for precipitation are given in Karger et al. (2021).

## 3. Evaluation

The evaluation of the downscaling from low (0.5°) to high (30 arcsec) resolution follows the evaluation approach outlined in Karger et al., (2021), and compares measurements at meteorological stations with data from both the low and the high spatial resolution. Since many observations at stations are already included in the W5E5 data due to the bias correction applied, we do not only evaluate the actual measurements at the stations, but rather focus on the difference between evaluation metrics achieved by the 0.5° data and the downscaled data. This will directly evaluate the downscaling, but not the forcing of the downscaling (see: Karger et al. 2021). We use two observational datasets, GHCN-D (Global Historical Climatology Network Daily) and GEBA (Global Energy Balance Archive), as references for the evaluation. The evaluation is performed at daily, seasonal, and long term climatological normals. The comparison to the station data is global, whereas the comparison to the dynamically downscaled data is constrained to the United States, where model output as well as a dense network of observational station data are available.

### 3.1 Evaluation datasets

To evaluate the performance of the downscaling algorithm we compute several test statistics at the original 0.5° resolution of the W5E5 data, and the downscaled data at 30 arcsec from CHELSA-W5E5. We use observations at meteorological stations (Table 1) and compare those to W5E5 and CHELSA-W5E5 data from the corresponding 0.5° and 30 arcsec grid cells, respectively, to assess the value added by the downscaling.

**Table 1:** Overview of the datasets used for evaluation, the variables contained, their temporal resolution and the number of stations used for the evaluation. *tas* = daily mean 2 m air temperature, *pr* = daily mean precipitation, *tasmax* = daily maximum 2 m air temperature, *tasmin* = daily minimum 2 m air temperature, and *rsds* = shortwave downwelling radiation.

| variable | dataset | number of stations used | temporal resolution | Reference |
|----------|---------|--------------------------|---------------------|-----------|
| *tas* | GHCN-D | 9225 | daily | (Menne et al., 2018) |
| *tasmin* | GHCN-D | 24994 | daily | (Menne et al., 2018) |
| *tasmax* | GHCN-D | 25018 | daily | (Menne et al., 2018) |
| *pr* | GHCN-D | 76369 | daily | (Menne et al., 2018) |

| *rsds* | GEBA | 1104 | monthly | (Wild et al., 2017) |
| --- | --- | --- | --- | --- |

### 3.1.1 GHCN-D

For the evaluation of 2 m air temperatures and precipitation rates, we used observations at meteorological stations from the Global Historical Climatology Network Daily (GHCN-D) network. This dataset contains meteorological station-based measurements from global land areas. About two thirds of the observations are precipitation measurements only (Menne et al., 2018).

### 3.1.2 GEBA

The station data of the GHCN-D network does not include energy flux variables. Thus, for the validation of shortwave downwelling radiation we used the Global Energy Balance Archive (GEBA). This database is maintained by the Institute for Climate and Atmospheric Sciences (IAC) at ETH Zurich and consists of globally measured energy fluxes at the Earth's surface (Wild et al., 2017). Its first version was implemented in 1988, it has continuously been updated ever since and mainly been improved in terms of data availability, data access, and internet appearance (Wild et al., 2017). GEBA provides observations for 15 surface energy flux components. Shortwave radiation incident at Earth's surface (global radiation) is the most widely measured quantity available in GEBA. The various observations have been compiled to monthly mean surface energy flux data from various sources.

### 3.2 Evaluation using observations at meteorological stations

To show the improvement resulting from the downscaling from 0.5° to 30 arcsec we compared each variable from both CHELSA-W5E5 and W5E5 to observations from meteorological stations (Table 1). For each meteorological station, the value of the grid cell that contains the location of the station was extracted and evaluated using several evaluation metrics.

### 3.2.1 Evaluation metrics

Evaluation metrics include the bias, correlation coefficient, root mean squared error and mean absolute error. The correlation is calculated based on Pearson's correlation coefficient,

$$r = \frac{cov(x_{sim}, x_{obs})}{\sigma(x_{sim})\,\sigma(x_{obs})}, \tag{25}$$

where $x_{obs}$ represents the observed time series at a meteorological station $x_{sim}$ the downscaled timeseries, *cov* the covariance, and $\sigma$ the standard deviation. The root mean squared error (*rmse*) is defined as

$$rmse = \sqrt{\frac{1}{n}\left(\sum_{i=0}^{n}\left(x_{sim_i} - x_{obs_i}\right)^2\right)}, \tag{26}$$

where $n$ is the number of time steps of a timeseries. Furthermore, the mean absolute error (*mae*) was computed according to

$$mae = \frac{1}{n}\left(\sum_{i=0}^{n} |x_{sim_i} - x_{obs_i}|\right) \tag{27}$$

Finally, the relative bias was computed to investigate the average amount by which the observations are greater than the estimates of the model output data based on different resolutions by

$$bias = x_{obs_i} - x_{sim_i} \tag{28}$$

10   **3.2.2 Seasonal performance**

To investigate if the downscaling has a similar performance throughout the year, in a first step we aggregated the daily or, in the case of *rsds*, the monthly data to seasonal values, e.g., winter = December, January, February, spring = March, April, Mai, summer = June, July, August, autumn = September, October, November. Based on the seasonally aggregated means, Taylor diagrams were used to show the performance improvements based on correlations, standard deviation and root mean squared error. Additionally, we calculated the Pearson correlation coefficient and the absolute bias between daily modelled values of either CHELSA-W5E5 or W5E5 and daily observations from GHCN-D and aggregated them to monthly means to assess possible trends in these two performance metrics over time. In the case of *rsds* monthly means instead of daily values were used.

**3.2.3 Global and regional performance**

20   Further comparisons between observations from meteorological stations, W5E5 and CHELSA-W5E5 were done also at daily resolution (in case of *rsds* a monthly resolution was used), as well as for long term climatological normals. Additional analyses were carried out for North America (except for *rsds*), where both the density of meteorological stations and their quality is high. Both globally and for North America, several evaluation metrics were calculated (see section 3.2.1). The main focus was on the difference in bias between CHELSA-W5E5 and W5E5 as this difference is an indicator of the value added by the downscaling algorithm.

To compare the performance spatially, we calculated the Pearson correlation between either CHELSA-W5E5 or W5E5 using daily values from either model in comparison to GHCN-D values for all meteorological stations globally. We then calculated the difference in the Pearson correlation coefficient and took the mean of all stations within a 0.5° grid cell that overlapped with these stations.

30   **3.2.3 Evaluation at the extremes**

To evaluate the performance of the downscaling at the extremes of the temperatures and precipitation rates we defined extreme values based on quantiles over the entire time period 1979-2016. For extreme high temperatures we used the 95[th] percentile of *tasmax*. Extreme precipitation rates were defined as the 95th percentile precipitation rates on wet days (days with *pr* greater than 0.1 kg m$^{-2}$ day$^{-1}$), and for extreme cold days we used the 5[th] percentile

35   of *tasmin*,

### 3.2.4 Comparison with dynamically downscaled data

To compare the terrain based downscaling to a more complex and computationally demanding dynamical downscaling, our evaluation includes a comparison with a simulation of the Weather Research and Forecast Model (WRF) (Skamarock et al., 2019) for the historical climate of North America (Rasmussen and Liu, 2017). The simulation was performed over a 13-year period (October 2000 - September 2013) with boundary conditions from ERA-Interim, at a spatial resolution of 4 km. The comparison between WRF and CHELSA-W5E5 was conducted for the variables *tas* and *pr*.

### 4. Results

### 4.1. Evaluation using observations at meteorological stations

### 4.1.1 Seasonal performance

The correlation of both datasets with observations at meteorological stations is very high overall ($r > 0.9$) for all variables globally as well as for North America except for daily *pr*. In general, the downscaling decreased the *bias*, *rmse*, *mae*, and increases the correlation for all variables expect *rsds* (Fig. 2, Table 2). There is no obvious deviation during any of the four seasons for *tas, tasmax*, or *tasmin* and the downscaling seems to perform equally well (Fig. 2). For *pr* the performance of both W5E5 and CHELSA-W5E5 is slightly higher during the northern winter months, while for *rsds* it is higher during northern spring and summer (Fig. 2).

**Mean Daily 2m Air-Temperature**

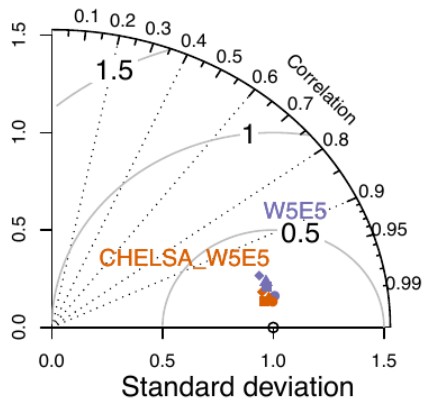

**Minimum Daily 2m Air-Temperature**

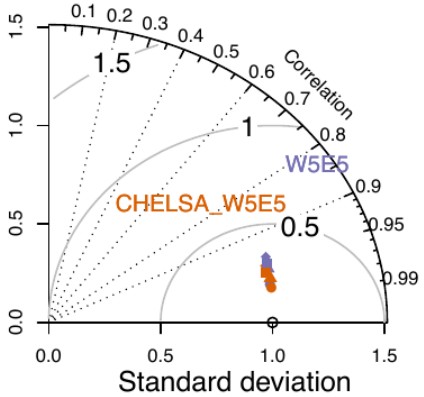

**Maximum Daily 2m Air-Temperature**

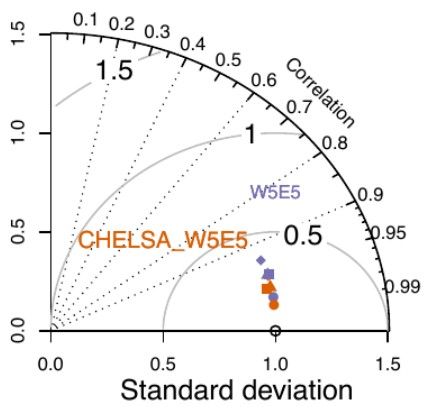

**Mean Daily Precipitation**

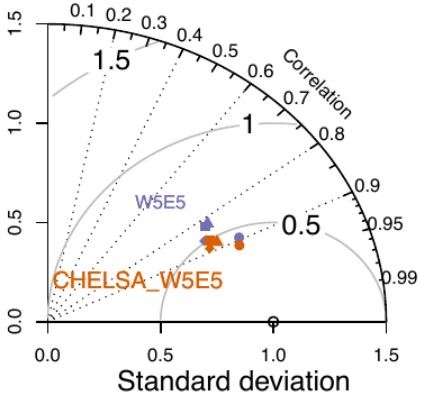

**Monthly Shortwave Downwelling Radiation**

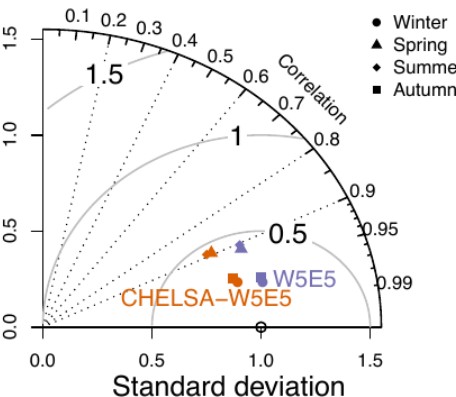

**Figure 2:** Seasonal performance based on a comparison of global long term seasonal means normals (1979-2016) of the global topographically downscaled high-resolution (30 arcsec, i.e. ~1 km) data (CHELSA-W5E5, orange) and the coarse (0.5°) original data (W5E5, violet) with GHCN-D for daily mean 2 m air temperature (*tas*), daily minimum 2 m air temperature (*tasmin*), daily maximum 2 m air temperature (*tasmax*), precipitation (*pr*), and shortwave downwelling radiation (*rsds*), based on monthly aggregated data. Values are shown separately for the four seasons: winter (DJF), spring (MAM), summer (JJA), autumn (SON). For the variables *tas*, *tasmin*, *tasmax*, and *pr*, the observational dataset GHCN-D was used for comparison. For *rsds*, the GEBA dataset was used.

### 4.1.2 Temporal performance

Both CHELSA-W5E5 and W5E5 do not show any significant trend in their performance when compared with observations at meteorological stations from GHCN-D (for *tas*, *tasmin*, *tasmax*, *pr*) or GEBA (for *rsds*) globally (Fig. 3). In general, the downscaled data shows a slightly higher Pearson correlation coefficient *r* with observations

5    than the coarse resolution W5E5 data, except for *rsds*. The overall pattern in the Pearson correlation coefficient *r* overall is also similar between CHELSA-W5E5 and W5E5 for all variables. The absolute *bias* is more variable compared to *r* with a generally lower bias but similar patterns for *pr*, *tas*, and *rsds* (Fig. 3), a mixed pattern of a higher *absolute bias* in the 1980's, and after 2000 with an otherwise lower *absolute bias*, and an higher absolute bias throughout for *tasmin*.

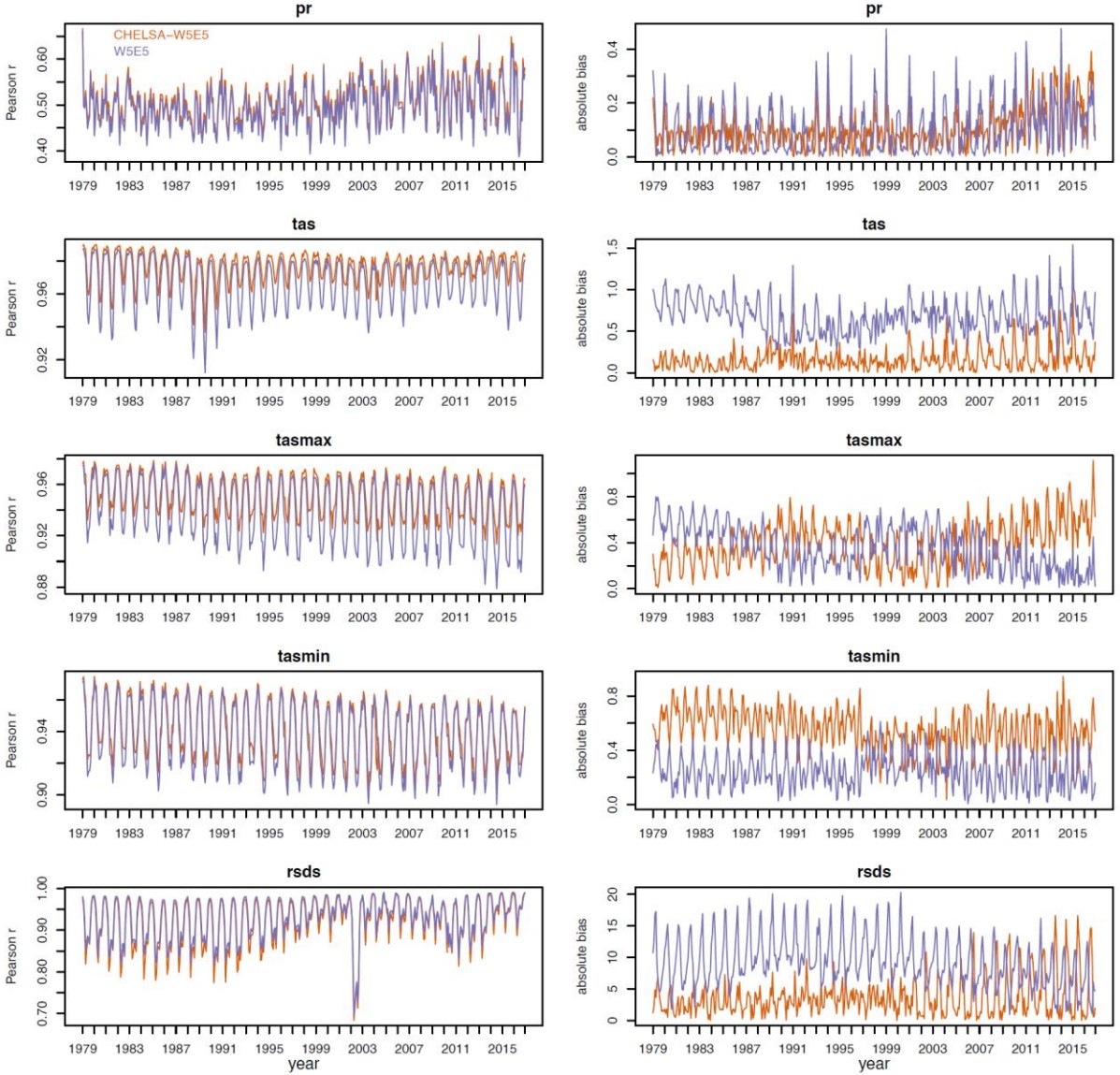

**Figure 3:** Mean daily Pearson correlation *r* and absolute bias between GHCN-D stations and downscaled CHELSA-W5E5 (orange), as well as W5E5 (purple) calculated for each month from 1979-2016 separately for daily mean 2 m air temperature

15    (*tas*), daily minimum 2 m air temperature (*tasmin*), daily maximum 2 m air temperature (*tasmax*), precipitation (*pr*), and shortwave downwelling radiation (*rsds*) globally.

### 4.1.2 Global and regional performance

For *tas, tasmax*, and *pr*, all error metrics (*bias, mae, rmse*) decrease after downscaling and the correlation coefficient increases (Fig. 4, Table 2). For *rsds* the bias is substantially reduced in the downscaled data, but the correlation coefficient is also slightly reduced (Fig. 4, Table 2). The lower correlation with yet a smaller bias seems to be driven by a systematic deviation of the downscaled *rsds* in areas with high *rsds* (Fig. 2). For *tasmin*, the pattern is opposite to *rsds*, i.e., the correlation coefficient increases after downscaling but the bias increases (Table 2). This pattern for *tasmin* and *rsds* is even more pronounced when only stations in North America are used (Table 3). The reduction in bias and increase in correlations of air temperatures due to the downscaling to 30 arcsec is highest in topographically heterogeneous terrain (Fig. 4), such as the western parts of North America whereas the topographic downscaling hardly added value in flat terrain (Fig. 5). Bias reduction and an increase in precipitation for precipitation is also highest in topographically complex terrain globally (Fig. 4), but considerable in flat terrain as well (Fig. 4, 5).

In regions with high-quality meteorological stations, such as the continental United States, the strong reduction in bias after downscaling in topographically complex terrain is also visible for *tas*, *tasmax* and *tasmin* (Fig. 6). For *tasmin*, in the middle of the Rocky Mountains, the bias in the downscaled data is significantly higher than for *tas* and *tasmax*, both of which show less bias in the downscaled data in this region. *tasmax* and *tasmin* both show higher bias in the downscaled data over flat terrain. For *pr*, the patterns are similar to those for air-temperatures, except that the bias is often lower over flat terrain (Fig. 6).

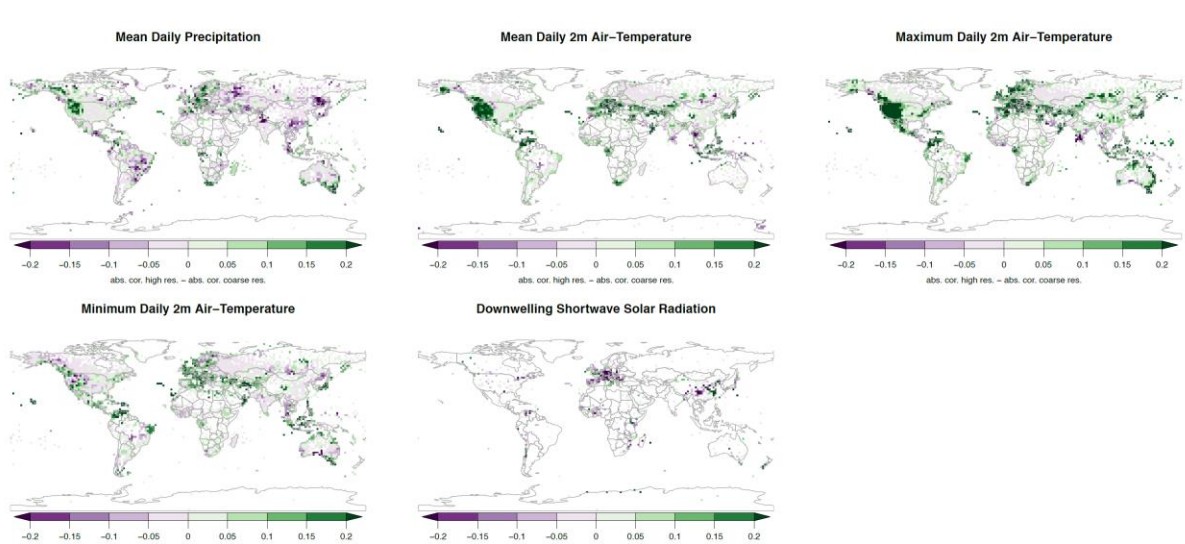

**Figure 4:** Mean differences in Pearson's correlation coefficient r between daily observations at meteorological stations CHELSA-W5E5 and W5E5 over the period 1979.2016. Negative values (violet) indicate areas in which a decrease in the correlation between observations after downscaling is observed, while positive values (green) indicate areas with an increase in the correlation coefficient (green). Observations are based on GHCN-D for daily mean 2 m air temperature (*tas*), daily minimum 2 m air temperature (*tasmin*), daily maximum 2 m air temperature (*tasmax*), precipitation (*pr*), and GEBA for shortwave downwelling radiation (*rsds*).

**Table 2:** Statistical scores from the comparison between CHELSA-W5E5 and W5E5, with observations from meteorological stations for all five variables (tas = daily mean 2 m air temperature, pr = daily mean precipitation, tasmax = daily maximum 2 m air temperature, tasmin = daily minimum 2 m air temperature, and rsds = shortwave downwelling radiation) globally. temp. res. = temporal resolution, bias=bias between a modelled value and a measurement at a specific timestep (temp. res.) at a specific station, sd_bias = standard deviation in bias, bias_re = reduction in bias (positive values indicate an increased performance), sd_bias_re = standard deviation in bias reduction, r = Pearson correlation coefficient, mae = mean absolute error, rmse = root mean squared error. Normals were calculated by averaging values over the entire observation period of a station between 1979-2016. Bias, sd_bias, bias_re, sd_bias_re, r, mae, and rmse have been based on comparisons of measurements between CHELSA-W5E5, W5E5, and observations at each station at each respective timestep (temp. res). Bold values in bias_re indicate an increase in performance due to the downscaling.

| model | variable | unit | temp. res. | bias | sd_bias | bias_re | sd_bias_re | r | mae | rmse |
|---|---|---|---|---|---|---|---|---|---|---|
| CHELSA-W5E5 | tas | K | daily | 0.053 | 2.369 | **0.378** | 1.429 | 0.984 | 1.601 | 2.369 |
| W5E5 | tas | K | daily | 0.660 | 2.755 | - | - | 0.979 | 1.978 | 2.833 |
| CHELSA-W5E5 | tasmin | K | daily | -0.548 | 2.996 | **0.080** | 1.404 | 0.966 | 2.197 | 3.046 |
| W5E5 | tasmin | K | daily | 0.247 | 3.123 | - | - | 0.963 | 2.276 | 3.132 |
| CHELSA-W5E5 | tasmax | K | daily | -0.386 | 2.949 | **0.288** | 1.415 | 0.972 | 2.096 | 2.974 |
| W5E5 | tasmax | K | daily | 0.334 | 3.283 | - | - | 0.965 | 2.384 | 3.300 |
| CHELSA-W5E5 | pr | kg m$^{-2}$day$^{-1}$ | daily | 0.004 | 0.707 | **0.008** | 0.162 | 0.511 | 0.244 | 0.707 |
| W5E5 | pr | kg m$^{-2}$day$^{-1}$ | daily | -0.004 | 0.733 | - | - | 0.499 | 0.252 | 0.733 |
| CHELSA-W5E5 | rsds | W m$^{-2}$ | monthly | 1.273 | 19.256 | **1.098** | 12.360 | 0.900 | 12.732 | 19.289 |
| W5E5 | rsds | W m$^{-2}$ | monthly | -10.329 | 18.731 | - | - | 0.914 | 13.830 | 21.382 |
| CHELSA-W5E5 | tas | K | normals | 0.086 | 1.217 | **0.510** | 1.235 | 0.990 | 0.830 | 1.220 |
| W5E5 | tas | K | normals | 0.729 | 1.812 | - | - | 0.980 | 1.340 | 1.953 |
| CHELSA-W5E5 | tasmin | K | normals | -0.564 | 1.636 | -0.015 | 1.192 | 0.980 | 1.282 | 1.731 |
| W5E5 | tasmin | K | normals | 0.220 | 1.824 | - | - | 0.970 | 1.268 | 1.837 |
| CHELSA-W5E5 | tasmax | K | normals | -0.408 | 1.701 | **0.289** | 1.221 | 0.980 | 1.057 | 1.749 |
| W5E5 | tasmax | K | normals | 0.298 | 2.192 | - | - | 0.960 | 1.346 | 2.212 |
| CHELSA-W5E5 | pr | kg m$^{-2}$day$^{-1}$ | normals | 0.027 | 0.550 | **0.015** | 0.284 | 0.900 | 0.326 | 0.551 |
| W5E5 | pr | kg m$^{-2}$day$^{-1}$ | normals | -0.039 | 0.575 | - | - | 0.900 | 0.342 | 0.576 |
| CHELSA-W5E5 | rsds | W m$^{-2}$ | normals | 0.696 | 23.628 | -0.866 | 13.997 | 0.958 | 16.360 | 23.638 |
| W5E5 | rsds | W m$^{-2}$ | normals | -9.550 | 21.751 | - | - | 0.963 | 15.494 | 23.755 |

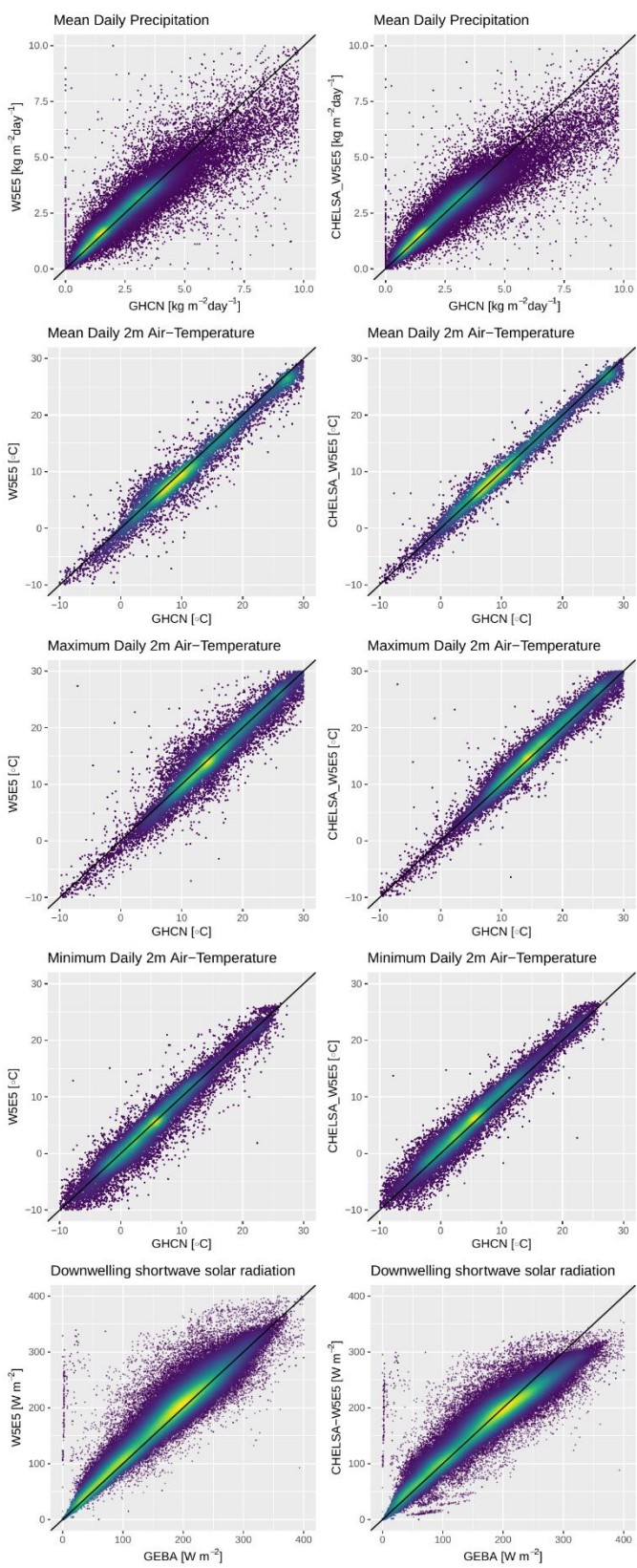

**Figure 5:** Scatterplots comparing long term mean observations from GHCN-D with values from W5E5 (left column, before downscaling), and CHELSA-W5E5 (right column, after downscaling). Each point represents the mean of all observations at a specific GHCN-D station in the period 1979-2016, except for downwelling shortwave solar radiation where each point represents a specific month.

**Table 3**. Statistical scores from the comparison between the two simulated datasets CHELSA-W5E5 and W5E5, and observations from GHCN-D stations in North America for all five variables (tas = daily mean 2 m air temperature, pr = daily mean precipitation, tasmax = daily maximum 2 m air temperature, tasmin = daily minimum 2 m air temperature, and rsds = shortwave downwelling radiation) globally. temp. res. = temporal resolution, bias=bias between a modelled value and a measurement at a specific timestep (temp. res.) at a specific station, sd_bias = standard deviation in bias, bias_re = reduction in bias (positive values indicate an increased performance), sd_bias_re = standard deviation in bias reduction, r = Pearson correlation coefficient, mae = mean absolute error, rmse = root mean squared error. Normals were calculated by averaging values over the entire observation period of a station between 1979-2016. Bias, sd_bias, bias_re, sd_bias_re, r, mae, and rmse have been based on comparisons of measurements between CHELSA-W5E5, W5E5, and observations at each station at each respective timestep (temp. res). Bold values in bias_re indicate an increase in performance due to the downscaling.

| model | variable | unit | temp. res. | bias | sd_bias | bias_re | sd_bias_re | r | mae | rmse |
|---|---|---|---|---|---|---|---|---|---|---|
| CHELSA_W5E5 | tas | K | daily | 0.062 | 2.992 | 0.562 | 1.585 | 0.964 | 2.071 | 2.993 |
| W5E5 | tas | K | daily | 0.176 | 3.411 | - | - | 0.952 | 2.459 | 3.415 |
| CHELSA_W5E5 | tasmax | K | daily | -0.741 | 3.234 | 0.163 | 1.464 | 0.965 | 2.393 | 3.318 |
| W5E5 | tasmax | K | daily | -0.005 | 3.566 | - | - | 0.957 | 2.618 | 3.566 |
| CHELSA_W5E5 | tasmin | K | daily | -0.691 | 3.125 | -0.134 | 1.428 | 0.959 | 2.319 | 3.200 |
| W5E5 | tasmin | K | daily | 0.089 | 3.206 | - | - | 0.957 | 2.334 | 3.207 |
| CHELSA_W5E5 | pr | kg m-2 day-1 | daily | -0.077 | 6.902 | 0.016 | 1.625 | 0.583 | 2.564 | 6.903 |
| W5E5 | pr | kg m-2 day-1 | daily | -0.128 | 7.186 | - | - | 0.569 | 2.643 | 7.187 |
| CHELSA_W5E5 | tas | K | normals | 0.108 | 1.277 | 0.562 | 1.249 | 0.979 | 0.883 | 1.281 |
| W5E5 | tas | K | normals | 0.520 | 1.947 | - | - | 0.950 | 1.445 | 2.015 |
| CHELSA_W5E5 | tasmax | K | normals | -0.749 | 1.661 | 0.163 | 1.246 | 0.970 | 1.138 | 1.822 |
| W5E5 | tasmax | K | normals | -0.031 | 2.199 | - | - | 0.948 | 1.301 | 2.200 |
| CHELSA_W5E5 | tasmin | K | normals | -0.668 | 1.744 | -0.134 | 1.182 | 0.960 | 1.415 | 1.867 |
| W5E5 | tasmin | K | normals | 0.098 | 1.844 | - | - | 0.955 | 1.281 | 1.847 |
| CHELSA_W5E5 | pr | kg m-2 day-1 | normals | 0.137 | 1.443 | 0.016 | 0.320 | 0.871 | 0.550 | 1.449 |
| W5E5 | pr | kg m-2 day-1 | normals | 0.087 | 1.433 | - | - | 0.867 | 0.566 | 1.436 |

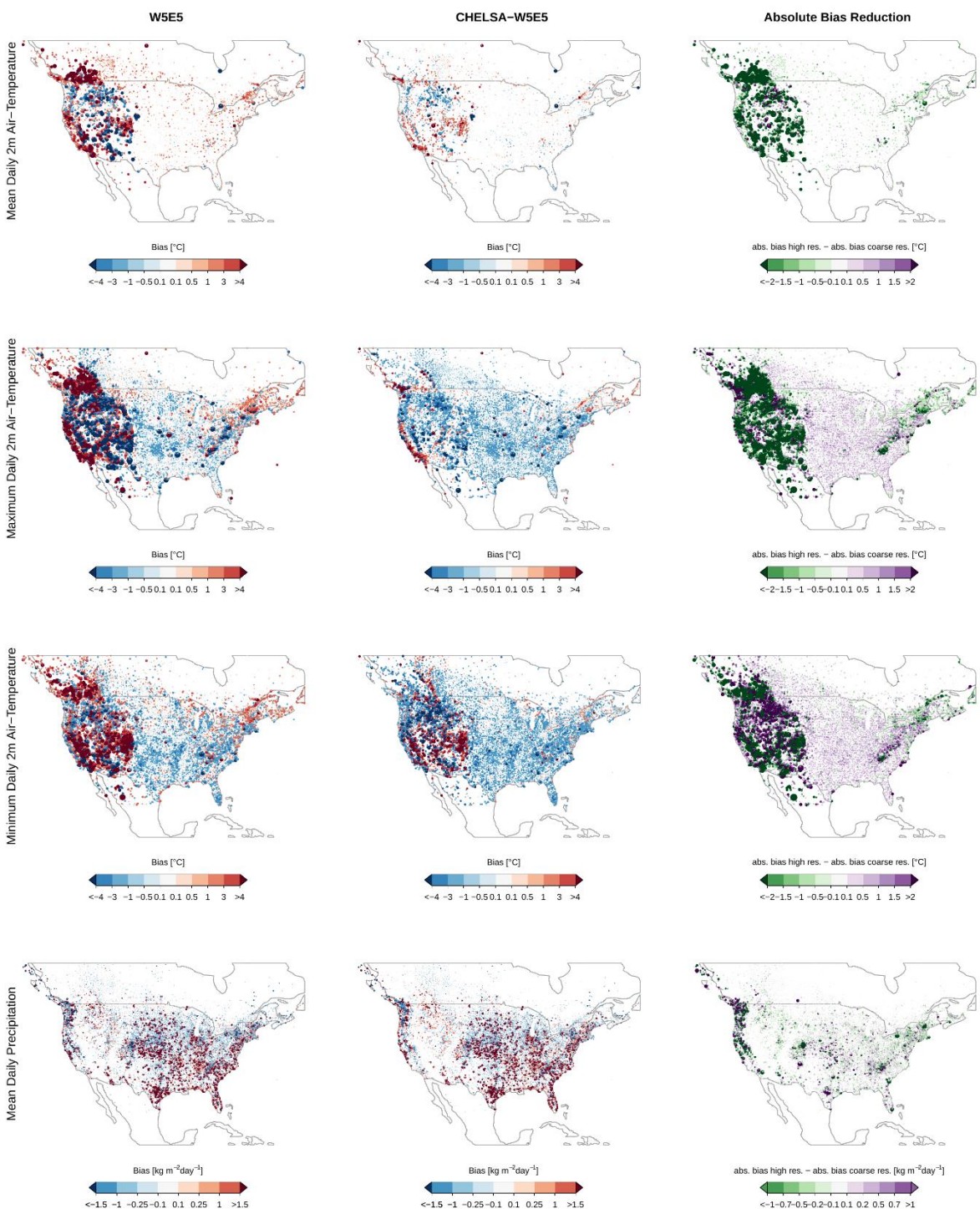

**Figure 6:** Mean bias of daily 2 m air temperatures and daily mean precipitation rates (from top to bottom) in North America averaged over the entire observational period of each station between 1979-2016. **Left:** Bias between W5E5 and observations at GHCN-D meteorological stations. **Middle:** Bias between the downscaled CHELSA-W5E5 and observations at GHCN-D meteorological observations. **Right:** Bias reduction at each of the stations as a result of the downscaling, i.e., the changes in absolute bias between the 0.5° W5E5 and 30 arcsec CHELSA-W5E5, with negative values indicating a bias reduction, and positive values indicating an increase in bias. The diameter of each dot scales with the absolute bias.

### 4.1.3 Extreme temperatures and precipitation

For extreme values such as the 95th percentile of daily maximum 2 m air temperature and the 5th percentile of daily minimum 2 m air temperature the bias reduction is again strongest in topographically complex terrain (Fig. 7, a, b). For extreme precipitation, the bias reduction is spatially not as coherent as for air temperature extremes and the bias can even increase with the downscaling. Generally, the downscaling shows a higher bias reduction in topographically complex terrain, while in flat terrain the downscaling actually introduces a bias in the extremes (Fig. 4, c).

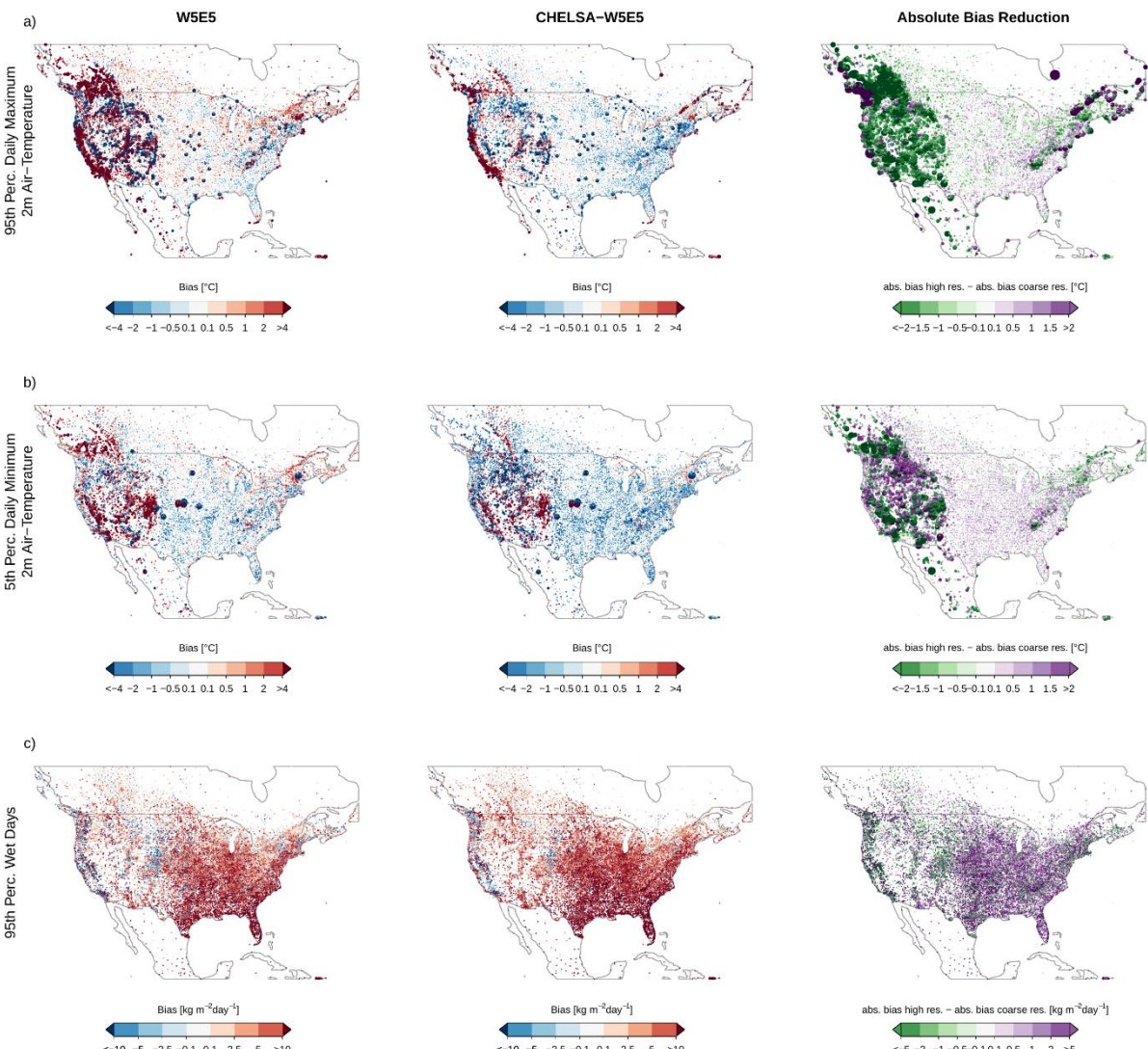

**Figure 7:** Mean bias of the extremes in maximum and minimum daily 2 m air temperatures and precipitation rates (from top to bottom) in North America averaged over the entire observational period of each station between 1979-2016. **Left:** Mean bias between W5E5 and observations from GHCN-D meteorological stations for extreme values in air temperature and precipitation. **Middle:** Bias between CHELSA-W5E5 and observations from GHCN-D meteorological stations for extreme values in air temperature and precipitation. **Right:** Absolute bias reduction after downscaling from 0.5° to 30 arcsec for extreme values in air temperature and precipitation, defined as the difference between the absolute bias of W5E5 and the absolute bias of CHELSA-W5E5.

### 4.2. Comparison with dynamically downscaled data

Both the downscaled air temperatures as well as precipitation rates from CHELSA-W5E5 and WRF show relatively high congruence with observations at meteorological stations (Fig. 8). Correlation rates are overall higher, and biases are lower for CHELSA-W5E5 than WRF when both models are compared to observations at GHCN-D stations over the same observational period (Table 5, Fig. 8). Correlations are almost similar for air temperatures, but slightly higher for CHELSA-W5E5 for precipitation compared to WRF (Fig. 8).

**Table 5**. Statistical scores from the comparison of CHELSA-W5E5 and WRF with observations from GHCN-D stations in North America daily mean 2 m air temperature (*tas*) and daily mean precipitation (*pr*). temp. res. = temporal resolution, bias=bias between a modelled value and a measurement at a specific timestep (temp. res.) at a specific station, sd_bias = standard deviation in bias, r = Pearson correlation coefficient, mae = mean absolute error, rmse = root mean squared error. Normals were calculated by averaging values over the entire observation period of a station between 200-2013. Bias, sd_bias, r, mae, and rmse have been based on comparisons of measurements between CHELSA-W5E5, WRF, and observations at each station at each respective timestep (temp. res). Bold values in bias_re indicate an increase in performance due to the downscaling.

| model | variable | temp. res. | bias | sd_bias | r | mae | rmse |
|---|---|---|---|---|---|---|---|
| CHELSA_W5E5 | tas | daily | 0.169 | 2.734 | 0.966 | 1.940 | 2.739 |
| WRF | tas | daily | 0.242 | 2.897 | 0.967 | 2.030 | 2.907 |
| CHELSA_W5E5 | pr | daily | -0.862 | 68.840 | 0.584 | 25.526 | 68.845 |
| WRF | pr | daily | 23.247 | 78.295 | 0.452 | 24.557 | 81.673 |
| CHELSA_W5E5 | tas | normals | 0.133 | 1.590 | 0.988 | 1.104 | 1.595 |
| WRF | tas | normals | -0.297 | 3.736 | 0.939 | 2.243 | 3.748 |
| CHELSA_W5E5 | pr | normals | 1.369 | 22.099 | 0.813 | 8.392 | 22.141 |
| WRF | pr | normals | 29.101 | 35.356 | 0.710 | 29.129 | 45.792 |

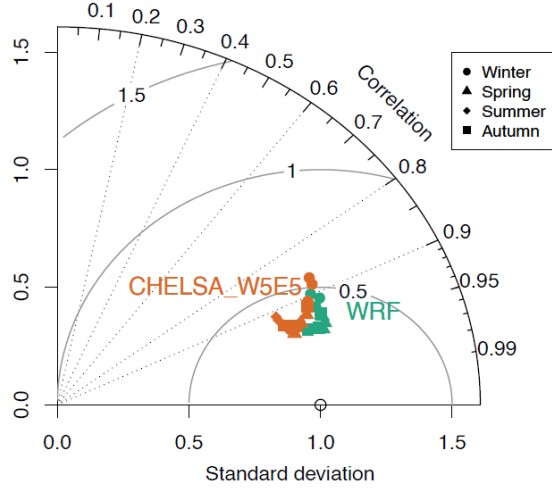

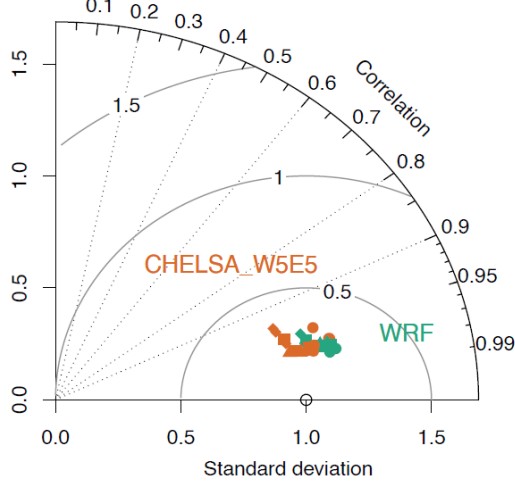

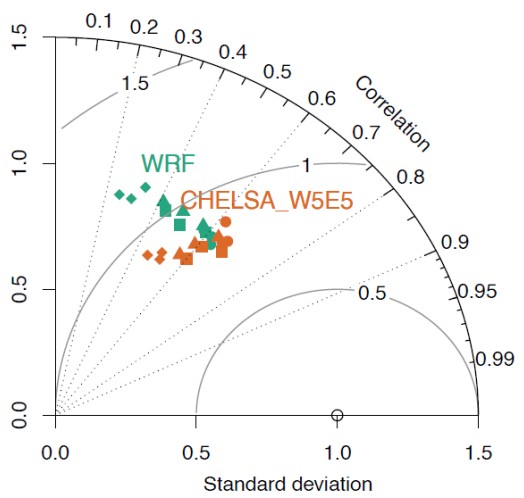

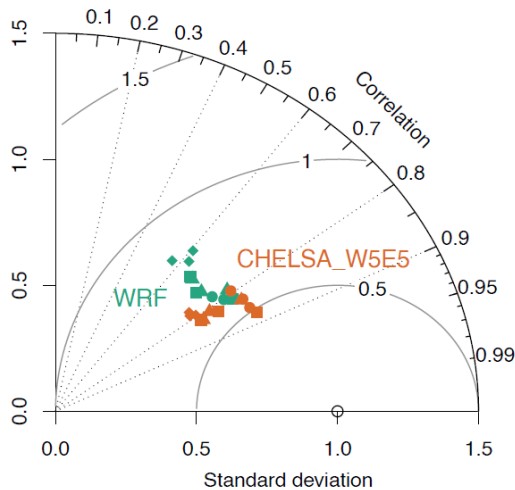

**Figure 8:** Performance based on a comparison of global long term monthly means of the topographically downscaled high-resolution (~1 km) data (CHELSA-W5E5, orange) with dynamically downscaled high-resolution (4 km) data (WRF, green) over North America, for the climatic variables daily mean 2 m air temperature and daily mean precipitation. The long-term means are shown separately for the four seasons; winter (DJF), spring (MAM), summer (JJA), autumn (SON).

## 5. Discussion

This paper shows that the CHELSA downscaling procedure generally increases the accuracy of the modelled air temperatures, precipitation rates, and downwelling shortwave solar radiation. While correlations between simulated and observed variables in the coarse 0.5° resolution W5E5 data are already generally greater than 0.9, the downscaling increases this correlation further and decreases the bias and errors of the data in most cases. Notably exceptions are *tasmin*, where the increase in correlation comes with an increase in the bias of the downscaled data, and *rsds*, where the reduction in bias comes with a decrease in the correlation with observations, specifically for high values of *rsds*.

There are no significant temporal trends in these two performance indicators (Pearson's *r*, *absolute bias*) visible. For the correlations between observations at meteorological stations both the coarse and the downscaled data show similar trends. This can be mainly attributed to the already good fit between the coarse data and the observations. Additionally, as the downscaling does not change the temporal pattern a similar correlation over time is expected. The absolute bias however, shows deviations between the downscaled and the coarse resolution data. While the bias for *pr*, *tas*, and *rsds* is generally lower in the downscaled data, *tasmax* shows a varying difference in bias over time, and *tasmin* a generally higher bias in the downscaled data. This trend might be attributed to the condition that mean daily temperature lapse rates are applied for all air-temperatures (*tas*, *tasmax*, and *tasmin*) equally, but under extreme conditions (*tasmax*, and *tasmin*) these lapse rates are not necessarily reflective of the observed conditions.

### 5.1 Air temperatures

The downscaling of the different air temperatures (*tas, tasmax, tasmin*) works best in topographically heterogeneous terrain, while its effect in flat terrain is much lower. This mainly comes from the relatively simple procedure applied that uses atmospheric temperature lapse rates, B-spline interpolations, and high-resolution orography alone to downscale air temperatures without any incorporation of, e.g., radiation budgets or air movements. Downscaling additionally improves the representation of temperature extremes, with absolute bias reductions exceeding those for mean temperatures.

The temperature downscaling does not use a full physical scheme as usually used in dynamical downscaling routines. Although the inclusion of additional effects other than the atmospheric lapse rate correction in a downscaling procedure would give more physically realistic estimates of air temperatures, the differences from such increase in complexity at very high resolutions are minimal in this case, as shown by the comparison with the numerically downscaled WRF data over North America. Dynamical downscaling however, comes at a large computational cost that makes it infeasible for global kilometre-scale application yet (Schär et al., 2019; Ban et al., 2021).

While overall, the performance of W5E5 and CHELSA-W5E5 is already high (r >= 0.9), the W5E5 data shows a lower fit with observations from GHCN-D during the spring and summer period. There are also limitation of the downscaling using mean daily lapse rates, especially for minimum 2m daily air-temperatures. The evaluation shows that downscaling *tasmin* with a mean daily temperature lapse rate as applied here can actually also increase the bias. In North America, this seems to happen especially in the high plateaus of the Rocky Mountains (Fig. 3), where minimum temperatures are usually caused during conditions of nocturnal inversions (Whiteman, 1982), causing positive temperature lapse rates with elevation. In this case the use of a mean daily temperature lapse rate

is not representative. Since the application of a different lapse rate for minimum daily 2m air-temperature and maximum daily 2m air-temperature could lead to higher minimum than maximum temperatures, this problem cannot be solved by running the CHELSA algorithm on a daily resolution, but only by increasing the temporal resolution and derive daily maximum- and minimum daily 2m air-temperatures from hourly downscaled air-temperatures

### 5.2 Precipitation

Downscaling also increases the correlation of precipitation with observations, although not to such a large degree as in the case of air temperatures. The coarse W5E5 data already has a high ($r$>0.9) correlation with observations, which is globally not much improved by the downscaling. However, the global comparison might be misleading here as the downscaling mainly affects precipitation rates at a very local scale where it has been shown to lead to large improvements (Karger et al. 2021). Topographic downscaling using the CHELSA v2.1 algorithm for precipitation rates has been shown to create long-term mean spatial patterns of precipitation rates that are extremely similar to those produced with dynamical downscaling using WRF over topographically complex terrain (Karger et al. 2021). A disadvantage of the presented precipitation downscaling is clearly that it cannot resolve convective precipitation, as only orographic effects are accounted for. While the mean bias in precipitation rates is generally decreased by the downscaling, the bias is larger during extreme precipitation events in topographically homogeneous terrain. These events are better captured by dynamically data using a dynamic model such as WRF at convection permitting resolutions.

### 5.3 Surface downwelling shortwave solar radiation

Surface downwelling shortwave solar radiation under clear-sky conditions is the only variable that is not directly downscaled but is fully mechanistically derived from terrain attributes. The algorithm for clear-sky solar radiation applied here captures terrain effects on solar radiation at very high spatial resolutions and has been shown to be effective in topographically complex terrain (Böhner and Antonic, 2009). Interpolations and direct downscaling are done on atmospheric cloud cover that is used to account for the amount of radiation which is absorbed and reflected by clouds. The high-resolution total cloud cover estimated by the algorithm has been shown to have monthly normals which correlate well with observations from GHCN-D ($r$=0.84, Brun et. al. 2022) even though the algorithm does not include convective cloud formation at kilometre-scale resolutions. While the bias is substantially reduced in the mid-range of *rsds* values, extreme high solar radiation shows stronger deviations from observations. This might be due to the relatively simple correction applied for *rsds* using cloud cover, or overestimates in the atmospheric scatter estimated with a bulk value of 80%. While it is unclear which part of the downscaling is responsible for the deviation at high *rsds* values, it shows where future developments of the downscaling should focus on and where clear limitations are visible.

### 5.4 Implications for applications

While the topographic downscaling increases the accuracy of the data, it most likely violates certain physical relationships, both due to the simplicity of the downscaling algorithm and since the five variables are downscaled independent from each other. These limitations are often encountered in univariate downscaling or bias correction

procedures (Zscheischler et al., 2019) and should be kept in mind when applying the output data of the downscaling in further analysis. Additionally, extreme values of *rsds* should be used with care and *tasmin* can show large deviations in areas with cold air pooling.

The data provided is additionally cropped by a land-sea mask that has been designed to include all 30 arcsec grid cells that overlap with a land mask, plus a buffer to account for potential spatial inaccuracies. This practically excludes all ocean surface areas. However, the algorithms applied here are solely forced by topography and if no topography is present, the downscaling is only done by a B-spline interpolation. Since this does not add information, we excluded all areas without topography to decrease the amount of data that needs to be stored.

## 6. Applications for impact modelling within ISIMIP

To test whether the improvements achieved by the downscaling, here shown as improved correlations and reduced biases compared to observed climate, also matter for impact modelling, the data will be further tested within the Inter-Sectoral Impact Model Intercomparison Project (ISIMIP). To this end, a range of impact models from different sectors (e.g. hydrological models, forest model or agricultural models) will be used to run at 1km and 0.5° resolution (and essentially a range of resolutions in between produced using the same approach as presented here for 1km) and compared to typical observational evaluation data for these impact models such as with ecosystem productivity data from eddy-covariance towers (Reyer et al., 2020) for forest models or discharge data for hydrological models (Huang et al., 2017; Liersch et al., 2020). Moreover, the CHELSA-W5E5 dataset will be employed to bias-adjust future climate projections in the upcoming ISIMIP phase 3 at high resolution to also allow regional applications at high spatial resolution that are still consistent with the wider ISIMIP framework.

## 7. Conclusions

In conclusion, we show that the evaluation of the effectiveness of the CHELSA downscaling procedure applied to W5E5 improves the accuracy of modelled air temperatures, precipitation rates, and downwelling shortwave solar radiation. The downscaling generally increased the correlation between simulated and observed variables and decreased bias and errors in most cases. However, exceptions were noted in the case of *tasmin* and *rsds*. The downscaling of air temperatures was found to work best in topographically heterogeneous terrain, with improvements in the representation of temperature extremes. The downscaling of precipitation rates was found to lead to large improvements at a very local scale, but it could not resolve convective precipitation. Additionally, the downscaling of surface downwelling shortwave solar radiation was found to be also effective in topographically complex terrain. Despite these improvements, there are still limitations connected to the downscaling procedure, including the use of mean daily lapse rates to downscale *tasmin*, which can actually increase the bias in the data, and the inability of the downscaling to capture convective precipitation, that should be taken into account when applying the data in climate impact studies.

## 7. Data Availability

The output of the CHELSA-W5E5 model is freely available under a CC0 1.0 Universal Public Domain Dedication (CC0 1.0) license. Dirk N. Karger, Stefan Lange, Chantal Hari, Christopher P. O. Reyer, Niklaus E. Zimmermann

(2022): CHELSA-W5E5 v1.0: W5E5 v1.0 downscaled with CHELSA v2.0. (https://doi.org/10.48364/ISIMIP.836809.3, Karger et al., 2022)

## 8. Code Availability

Source codes of the CHELSA model used for the downscaling are available here: https://gitlabext.wsl.ch/karger/chelsa_isimip3b_ba_1km.git

Source codes of the evaluation are available here: https://gitlabext.wsl.ch/harichan/chelsa_w5e5-validation

## 9. Author contribution

D.N.K., S.L., and C.P.O.R. conceived the study with input from K.F. and N.E.Z. D.N.K. developed and conducted the downscaling. O.C. developed the *rsds* algorithm. C.H. and D.N.K. conducted the validation with input from S.L. and C.P.O.R.. D.N.K. wrote the first version of the manuscript with input from all co-authors and all authors contributed significantly to further revisions.

## 10. Competing interests

The authors declare that they have no conflict of interest.

## 11. Acknowledgements

This paper is based upon work undertaken as part of COST Action CA 19139 PROCLIAS, supported by the COST Association (European Cooperation in Science and Technology - www.cost.eu ) and also benefitted from discussion within the Intersectoral Impact Model Intercomparison Project (ISIMIP). DNK & NEZ acknowledge funding from: the WSL internal grant exCHELSA, the 2019–2020 BiodivERsA joint call for research proposals, under the BiodivClim ERA-Net COFUND program, with the funding organisations Swiss National Science Foundation SNF (project: FeedBaCks, 193907), as well as the Swiss Data Science Center Project: SPEEDMIND and COMECO. DNK acknowledges funding to the ERA-Net BiodivERsA - Belmont Forum, with the national funder Swiss National Science Foundation (20BD21_184131), part of the 2018 Joint call BiodivERsA-Belmont Forum call (project 'FutureWeb'), the WSL internal grant ClimEx, and the Swiss National Science Foundation (Project Adohris, 205530).S.L. acknowledges funding from the German Research Foundation (DFG, Project No.: 427397136) and the German Federal Ministry of Education and Research (BMBF, Grant ID: 01LP1907A). We thank the GEBA data providers for making their data available. GEBA is co-funded by the Federal Office of Meteorology and Climatology MeteoSwiss within the framework of GCOS Switzerland. Funding from the EU Horizon 2020 research and innovation program under grant agreement 821010 (CASCADES) supported the work of C.P.O.R.

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
