# Peer review of "CHELSA-W5E5: Daily 1 km meteorological forcing data for climate impact studies"

_Earth System Science Data, 2022_

## Referee Comment (RC1)

This manuscript published the daily 1 km (30 arcsec) meteorological forcing data set. It obviously improves the resolution of existing data products. It is very meaningful for climate impact studies and lays a foundation for the related topics. However, some other problems in the manuscript are still concerned in the following:

1. In the results (Table 5), the proposed CHELSA-W5E5 was only compared with WRF. Could the authors compare their data sets with more data sets for a better validation?
2. The correlation with observations at meteorological stations is evaluated in North America. How are other regions?
3. The organization of this manuscript should be added to the end of the introduction.
4. A flow chart of the data set production is suggested to be shown. It is very important for the coming readers.
5. A section of "conclusions" is suggested.

---

## Author Response (AR1)

**Response to reviewer 1**

This manuscript published the daily 1 km (30 arcsec) meteorological forcing data set. It obviously improves the resolution of existing data products. It is very meaningful for climate impact studies and lays a foundation for the related topics. However, some other problems in the manuscript are still concerned in the following:

Thank you for taking time to review the manuscript. We have taken your comments into account and adjusted the manuscript accordingly.

1. In the results (Table 5), the proposed CHELSA-W5E5 was only compared with WRF. Could the authors compare their data sets with more data sets for a better validation?

Check other data variables:

2. The correlation with observations at meteorological stations is evaluated in North America. How are other regions?

Response: The initial idea of only using North America was that the quality of the stations in this region is high compared to other regions. We however see the point of that it might be interesting to also show other regions with the remark that the error in the stations might be different across the regions.

We therefore opted for the following approach:
1. We keep the focus on North America for the already existing figures due to the reliable observations and dense station network in these regions.

2. We added a figure showing the temporal performance of the dataset (see also the comment of reviewer 2). This is now Figure 3.

3. We added a global overview of the correlation as a new figure to show the global performance. This is now Figure 4.

We used figure instead of tables in this case as they would be easier to interpret and we would not need a many additional tables, or expand the existing table 2 even further, making it less readable.

3. The organization of this manuscript should be added to the end of the introduction.

Response: We added a short organization of the manuscript at the end of the introduction to guide the reader through the remaining text. It says:

"Here we describe the CHELSA downscaling procedure applied to W5E5 and evaluate its performance in improving the accuracy of modelled air-temperatures, precipitation rates, and downwelling shortwave solar radiation. We give a description on the input data as well as the a detailed description of the downscaling procedure applied, which includes the downscaling of near-surface air temperature (tas, tasmax, tasmin), surface downwelling shortwave radiation (rsds), and precipitation (pr). We evaluate our results using observations at meteorological stations, and analyse the performance of

the downscaling globally, regionally, and seasonally, as well as at the extremes and additionally compare our results with dynamically downscaled data."

4. A flow chart of the data set production is suggested to be shown. It is very important for the coming readers.

Response: Thank you for the suggestion. We now added a flowchart highlighting the different steps of the algorithm as Figure 1.

5. A section of "conclusions" is suggested

Response: We now include a conclusion paragraph. It says:

"In conclusion, we show that the evaluation of the effectiveness of the CHELSA downscaling procedure applied to W5E5 improves the accuracy of modelled air temperatures, precipitation rates, and downwelling shortwave solar radiation. The downscaling generally increased the correlation between simulated and observed variables and decreased bias and errors in most cases. However, exceptions were noted in the case of tasmin and rsds. The downscaling of air temperatures was found to work best in topographically heterogeneous terrain, with improvements in the representation of temperature extremes. The downscaling of precipitation rates was found to lead to large improvements at a very local scale, but it could not resolve convective precipitation. Additionally, the downscaling of surface downwelling shortwave solar radiation was found to be also effective in topographically complex terrain. Despite these improvements, there are still limitations connected to the downscaling procedure, including the use of mean daily lapse rates to downscale tasmin, which can actually increase the bias in the data, and the inability of the downscaling to capture convective precipitation, that should be taken into account when applying the data in climate impact studies."

**Response to reviewer 2**

This paper developed a new long-term daily 1 km meteorological forcing dataset for air-temperature, precipitant and shortwave solar radiation, using the CHELSA topographic downscaling algorithm. This work is useful for climate impact studies. However, there are several things that need to be addressed before the paper can be accepted. The authors should go over the comments that I listed below and carefully address them.

Thank you for taking time to review the manuscript. We have taken your comments into account and adjusted the manuscript accordingly.

1. As shown in Table 2 and Figure 2, the performance of the CHELSA-W5E5 generally outperform the W5E5 for air temperature, precipitation, and downward shortwave radiation. Is the performance of the CHELSA-W5E5 varied with different regions of the world? I suggest more description and justification of the spatial-temporal accuracy of this dataset.

Response:   The initial idea of only using North America was that the quality of the stations in this region is high compared to other regions. We however see the point of that it might be interesting to also show other regions with the remark that the error in the stations might be different across the regions.

We therefore opted for the following approach:

1.   We keep the focus on North America for the already existing figures due to the reliable observations and dense station network in these regions.

2.   We added a figure showing the temporal performance of the dataset (see also the comment of reviewer 2). This is now Figure 3.

3.   We added a global overview of the correlation as a new figure to show the global performance. This is now Figure 4.

2. In Figure 3, for example in the sub-graph "Absolute Bias Reduction", it is hardly to identify the negative and positive values in the eastern North American. The color bar should be changes.

Response:   The color is actually a complementary contrast (green-violet), so it should be easy to distinguish. We chose this color over e.g. red-blue, as this one is used already for the bias. Different variables should have different colors in this case. However, the points also scale with the absolute value so that they become tiny if the difference in bias is tiny. We played around with different versions of this, but could not come up with a better visible one. We kept it therefore.

3. As shown in Figure 3 and 4, the obvious improvement of the CHELSA-W5E5 is observed in the western North American with complex terrain. For the eastern North American, does the CHELSA-W5E5 show the worse performance compared to W5E5? I suggest that the authors also give this information.

Response:   This is now included in the text in the following (underlined):

"In regions with high-quality meteorological stations, such as the continental United States, the strong reduction in bias after downscaling in topographically complex terrain is also visible for tas, tasmax and tasmin (Fig. 6). For tasmin, in the middle of the Rocky Mountains, the bias in the downscaled data is significantly higher than for tas and tasmax, both of which show less bias in the downscaled data in this region. tasmax and tasmin both show higher bias in the downscaled data over flat terrain. For pr, the patterns are similar to those for air-temperatures, except that the bias is often lower over flat terrain (Fig. 6)."

Minor comments

1. Page 14 line 15 "Downwelling shortwave solar radiation" can be changed to "downwelling shortwave solar radiation".

Response:   Changed everywhere

2. Page 19 line 14 "Then" should be changed to "than".

Response:   Changed

3. Page 20 Line 5 The color represented WRF data should be added in the title of Figure 5.

Response:   Changed